# Automated Microbial Library Generation Using the Bioinformatics Platform IDBac

**DOI:** 10.3390/molecules27072038

**Published:** 2022-03-22

**Authors:** Chase M. Clark, Linh Nguyen, Van Cuong Pham, Laura M. Sanchez, Brian T. Murphy

**Affiliations:** 1Department of Pharmaceutical Sciences, Center for Biomolecular Sciences, College of Pharmacy, University of Illinois at Chicago, 833 S. Wood St., Chicago, IL 60612, USA; cmclark8@wisc.edu; 2Institute of Marine Biochemistry, Vietnam Academy of Science and Technology, Nghiado, Caugiay, Hanoi 10000, Vietnam; nguyenthuylinhorc@gmail.com (L.N.); phamvc@imbc.vast.vn (V.C.P.); 3Department of Chemistry and Biochemistry, University of California Santa Cruz, 1156 High Street, Santa Cruz, CA 95064, USA; lmsanche@ucsc.edu

**Keywords:** IDBac, MALDI, bioinformatics, drug discovery, microorganisms, natural products

## Abstract

Libraries of microorganisms have served as a cornerstone of therapeutic drug discovery, though the continued re-isolation of known natural product chemical entities has remained a significant obstacle to discovery efforts. A major contributing factor to this redundancy is the duplication of bacterial taxa in a library, which can be mitigated through the use of a variety of DNA sequencing strategies and/or mass spectrometry-informed bioinformatics platforms so that the library is created with minimal phylogenetic, and thus minimal natural product overlap. IDBac is a MALDI-TOF mass spectrometry-based bioinformatics platform used to assess overlap within collections of environmental bacterial isolates. It allows environmental isolate redundancy to be reduced while considering both phylogeny and natural product production. However, manually selecting isolates for addition to a library during this process was time intensive and left to the researcher’s discretion. Here, we developed an algorithm that automates the prioritization of hundreds to thousands of environmental microorganisms in IDBac. The algorithm performs iterative reduction of natural product mass feature overlap within groups of isolates that share high homology of protein mass features. Employing this automation serves to minimize human bias and greatly increase efficiency in the microbial strain prioritization process.

## 1. Introduction

Recently, several steps in the microbial drug discovery pipeline have undergone significant advances including the analytical detection of natural products (NPs) [1,2], bioinformatic-driven NP dereplication [3], NP structure elucidation [4], genome mining [5], and biological activity screening [6,7]. However, the process of selecting culturable bacteria for addition to a strain library has seen relatively few widely applied conceptual changes since the 1940s [8]. To date, culturable isolates entering into a bacterial library are often selected based on common morphological phenotypes or 16S rRNA gene sequencing, which do not fully account for NP production [9,10,11,12]. These practices have simultaneously resulted in significant phylogenetic and NP overlap in strain libraries while failing to capture subtle differences in NP capacity between highly similar strains [13,14]. This results in a major obstacle facing microbial NP drug discovery efforts—the consistent re-discovery of known NPs [15,16].

To address this obstacle, we previously developed IDBac, an informatics pipeline that facilitates the data-driven prioritization of environmental bacterial isolates based on matrix-assisted laser desorption/ionization time-of-flight mass spectrometry (MALDI-TOF MS) analyses of cell material taken directly from a bacterial colony on a Petri dish [17,18]. In short, IDBac clusters spectra that represent ionized small proteins (2000–15,000 Daltons) of each isolate into a dendrogram, resulting in the creation of pseudo-phylogenetic groups. The program then allows users to assess the degree of NP overlap within each group of isolates through visualization of metabolite association networks (MANs, 200–2000 Daltons). Both datasets are acquired successively from a single sample preparation, and evaluation of both pseudo-phylogenetic and NP data affords researchers an opportunity to create a diverse microbial library while incorporating fewer environmental isolates. IDBac has the potential to significantly reduce costs associated with downstream library-based discovery, and towards community adoption; as of 1 March 2022, IDBac has been downloaded 1137 times from 36 countries.

Despite the efficiency of this pipeline, manual inspection of MANs was still required to determine areas of significant phylogenetic and NP overlap between isolates, resulting in potential researcher-introduced bias in the informatics pipeline [17]. This was particularly apparent when analyzing data from large sample collection expeditions. Typically, greater than 30 collected environmental samples plated on multiple media types resulted in up to thousands of environmental isolates to be processed and analyzed [19]. To mitigate potential bias, we designed and employed an algorithm to automate the prioritization of environmental bacterial isolates within each pseudo-phylogenetic group in the dendrogram. The algorithm seeks to iteratively select isolates until a user-specified percentage of unique *m*/*z* features is accumulated within each MAN. The result is the automated creation of a library of custom-size, maximized to capture unique *m*/*z* features. This process removes a degree of bias that accompanies manual library curation and serves to diminish the probability for human error while significantly increasing the speed and efficiency of the isolate selection process.

## 2. Results and Discussion

### 2.1. Collection and Processing of Environmental Samples

In collaboration with partners from the Vietnam Academy of Science and Technology and with the appropriate permission and permits, 130 marine samples (sediments, sponges, algae) were collected from several locations around the island of Phu Quoc in Southern Vietnam using SCUBA. Samples were plated on eight different media types to maximize bacterial diversity. After an incubation period, all distinguishable bacterial colonies were isolated and purified. A total of 819 bacterial isolates were purified and MALDI-TOF MS analysis was carried out using three technical replicates per isolate, acquiring data in protein and NP regions, as previously described [18,19]. The resulting data were processed in IDBac to generate a dendrogram of 819 isolates (Figure 1), grouped by protein spectrum similarity. An arbitrary cut-height designed to approximate species-level phylogenetic branching points was applied (see red line in Figure 1) resulting in the generation of 83 pseudo-phylogenetic subgroups (and therefore 83 MANs), which can be evaluated for NP overlap.

Since each MAN is composed of closely related isolates, analysis of *m*/*z* overlap within the MANs and subsequent selection of isolates with minimally overlapping mass features is critical to the creation of a streamlined bacterial library (see Figure 1b for an example MAN). However, this process relies on the judgement of an individual researcher as to what constitutes significant feature overlap, which can jeopardize the quality of a library due to varying levels of user bias. To automate and thus streamline this process as well as mitigate user-to-user variation, an algorithm was created and embedded into the IDBac code.

### 2.2. Application of the Isolate Prioritization Algorithm to a Collection of Unknown Environmental Bacterial Isolates

A prioritization function was added to the IDBac R package (see Data Availability Statement). The function requires a pseudo-phylogenetic dendrogram as input (created using IDBac) and a connection to an IDBac database (created by the IDBac software). The user is required to select the number of pseudo-phylogenetic groups that will be analyzed, either by “cutting” the dendrogram at an arbitrary height or by defining the desired number of groups. The user then defines a threshold percentage of unique small molecule mass features to be kept within each pseudo-phylogenetic group. A small threshold will result in fewer chosen isolates but a higher number of missed mass features while a large threshold will result in a higher number of chosen isolates and lower number of missed mass features. While for this study we chose an arbitrary 75% threshold, we would suggest this number be adjusted, per dataset, until the number of chosen isolates represents the workload a laboratory can handle in downstream studies. The algorithm then proceeds through each pseudo-phylogenetic group, identifies the isolate with the most NP mass features, selects and removes that isolate and its mass features from the group, then iterates this process until the threshold percentage of total mass features within the group is reached.

The automated prioritization algorithm was applied to the MALDI-TOF MS dataset generated for the 819 bacterial isolates. Using a threshold defined to capture 75% of the mass features, the algorithm selected 189 isolates from the 83 pseudo-phylogenetic subgroups. These 189 isolates represented a 77% reduction from the total number of isolates collected from the bacterial diversity plates. The algorithm achieved similar performance when compared with two researchers who were assigned to create a library from the same data set, with instructions to “minimize the number of isolates in the library while capturing approximately 75% of the mass features” (see Appendix A). Using manual curation, Researchers 1 and 2 achieved a 71% and 64% reduction in total number of isolates collected, finishing with a total of 236 and 293 isolates, respectively. While the algorithm captured similar chemical space, it was able to do so using 47 and 104 fewer isolates than Researchers 1 and 2, respectively. This reduction is significant given the high downstream costs of processing these additional isolates through the traditional pipeline of fermentation, extraction, fraction library generation, biological screening, and lead dereplication. Further, Researchers 1 and 2 each required approximately four hours to prioritize isolates from the MANs, while the automated process required only seconds.

The 819 isolates produced 6624 distinct mass features distributed across 83 MANs. It is important to note that multiple mass features can represent a singular chemical structure/entity (see further discussion of isotopologues and adducts in “Study limitations and perspectives”). An example comparison of manual and automated outputs from one pseudo-phylogenetic group is shown in Figure 2. The MANs of Group 8 depict the number of isolates/mass features retained through the library creation process; this was repeated for each of the 82 remaining groups. Both manual and automatic prioritization methods were successful at capturing the majority of *m*/*z* features present across the collection: 5455 overlapping features that represented an estimated 82% of the chemical space detected by the MALDI-TOF MS (Figure 3). The algorithm’s average across the 83 groups was higher than 75% because it is designed to select isolates continuously from a group until a minimum threshold percent of mass features is accumulated. Because mass features do not accumulate one-by-one but rather by their presence in iteratively chosen isolates, the accumulated mass features can be greater than the input threshold. Regardless, a major advantage of automating the isolate selection process is that the user is guaranteed to reach a threshold mass retention within each pseudo-phylogenetic group.

### 2.3. Study Limitations and Perspectives

For nearly eight decades the process of microbial library generation and curation has been labor intensive and required a significant effort and input from experts, who employed either morphology-guided colony selection from diversity plates, 16S rRNA sequencing of isolates, or both in order to create an isolate library for natural product discovery [8,9,10]. The current study represents an advance toward an automated microbial library creation pipeline, though some challenges remain. In its current form the algorithm retains the limitations of IDBac. Most relevant is a lack of deisotoping MALDI-TOF MS spectra (i.e., isotopologues are not collapsed into a single mass feature) and no dimensionality to allow for separation of isobars. Future versions of the prioritization algorithm may consider adding awareness of isotopologues and molecular analogues to allow for more high mass accuracy feature selection. Finally, within IDBac, the choice of where to ‘cut’ a dendrogram to create pseudo-phylogenetic groups remains manual, as automating this step remains a longstanding issue in the bacterial MALDI-TOF MS [20,21] field largely due to a lack of reference spectra from which to train more sophisticated computational models. With the availability of accurate, automated clustering of protein spectra, it is conceivable that the entire front end library generation process could be automated—robotic colony isolation and plating onto a MALDI target plate [22], MALDI-TOF MS data acquisition and processing, isolate prioritization, and isolate distribution into a library.

## 3. Conclusions

In the current study, we developed an algorithm that works within the existing bioinformatics platform IDBac to automate the prioritization of hundreds to thousands of environmental microorganisms based on MALDI-TOF MS protein and NP spectra. It does so through the iterative reduction of NP mass feature overlap within groups of isolates that share high homology of protein mass features. Employing this automation serves to minimize human bias and greatly increase efficiency during a major front end bottleneck in the drug discovery process. When applied to 819 unknown environmental bacterial isolates clustered into 83 pseudo-phylogenetic groups, the algorithm selected 189 isolates for retention, discarding 630 isolates. The 189 isolates harbored the majority of small molecule *m*/*z* features present across the collection, an estimated 82% of the chemical space detected by the MALDI-TOF MS. The algorithm achieved similar performance when compared with two researchers who were assigned to create a library from the same data set, and did so in seconds, compared to multiple hours required for manual curation.

## 4. Experimental Section

### Sample Collection and Processing MALDI-TOF MS

In October of 2018, a total of 130 sediment, sponge, and algae samples were collected from 18 locations in Vietnam and processed immediately following collection. Samples were collected under a long-standing partnership between UIC and the Institute of Marine Biochemistry at VAST, and with the appropriate permits and benefit sharing agreements. Samples were plated on SWA, SCA, M1, A1, PDA, ISP1, ISP2, and NZSG nutrient media (see Supplementary Material for recipes). After incubating up to 60 days, all distinguishable colonies were isolated and purified onto 60 mm Petri dishes containing A1 marine media. A total of 819 marine bacterial isolates were cultivated on high nutrient, marine media and incubated at room temperature for seven days. Following incubation, a small portion of each isolate was transferred to a 384-spot MALDI target plate (Bruker Daltonics, Billerica, MA, USA) in three technical replicates using a sterile toothpick. Then, 1 μL of 70% MS grade formic acid and 1 μL of MALDI matrix (10 mg/mL alpha-cyano-4-hydroxycinnamic acid prepared in 50% acetonitrile, 47.5% water and 2.5% trifluoroacetic acid) was added to each sample spot. MALDI-TOF MS analysis was performed using an Autoflex Speed LRF mass spectrometer (Bruker Daltonics) equipped with a Smartbeam-II laser (355 nm). Further MALDI-TOF MS settings and procedures can be found in our previous IDBac publications [17,19] and detailed protocol [18].

## Figures and Tables

**Figure 1 molecules-27-02038-f001:**
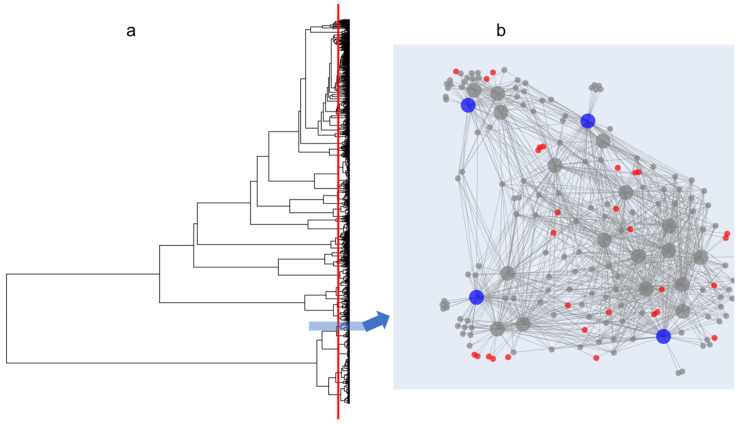
Dendrogram (**a**) created from 819 bacterial isolates (2000–15,000 Da; protein spectra run in triplicate for each isolate). An example Metabolite Association Network (MAN) is displayed in (**b**), where large circles represent bacterial isolates that are connected to smaller circles representing *m*/*z* features. Colored circles represent isolates and mass features chosen by the algorithm, discussed further in Figure 2. The pseudo-phylogenetic grouping of these (**b**) isolates is depicted by the blue box on the dendrogram.

**Figure 2 molecules-27-02038-f002:**
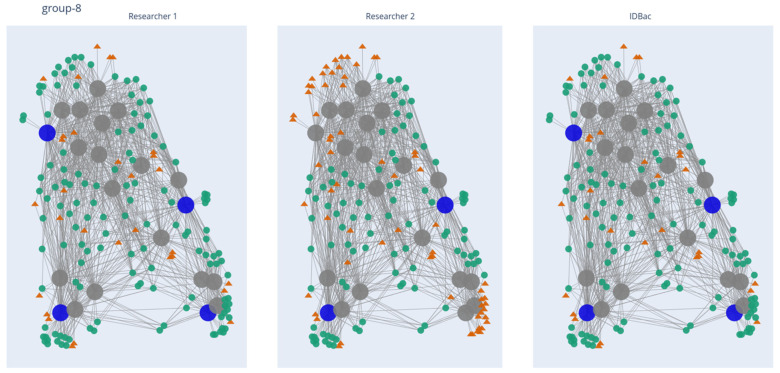
MAN of pseudo-phylogenetic Group 8, depicting the NP coverage resulting from manual and automatic prioritization processes. Large nodes represent bacterial isolates and small nodes represent mass features. Large blue nodes are isolates selected to be added to the library, while small green circles represent mass features “captured” by the selected isolates. Large grey nodes are isolates not selected, while small orange triangles are mass features “missed” by the selected isolates. Links to the other 82 groups may be found in Section Data Availability Statement.

**Figure 3 molecules-27-02038-f003:**
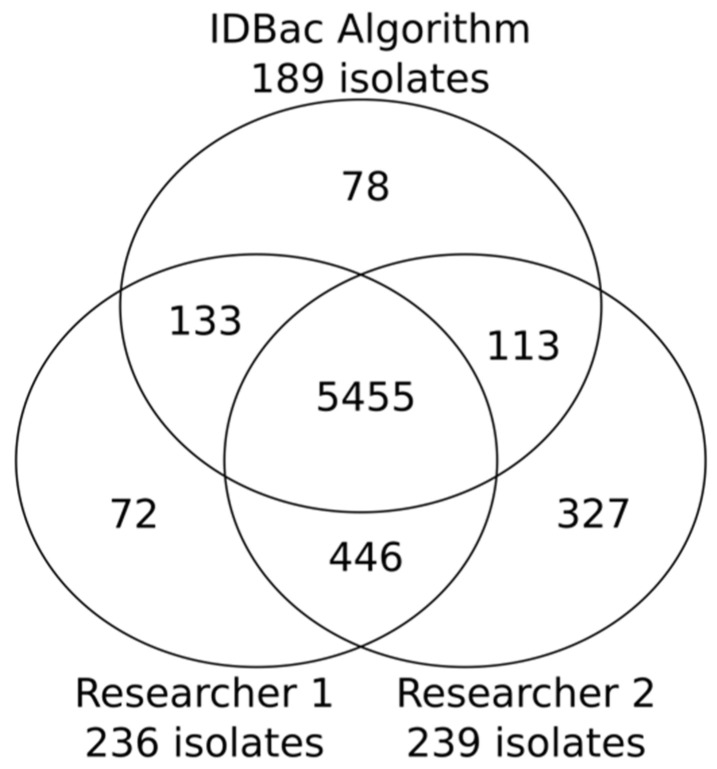
Venn diagram showing the number of mass features “captured” by the two researchers and the IDBac algorithm; 82% were captured by all three, though the algorithm did so with 47 and 104 fewer isolates than Researchers 1 and 2, respectively. The total isolates recovered from each group: Researcher 1: 236; Researcher 2: 293; the IDBac algorithm: 189.

## Data Availability

All data (including MANs), code, and docker instructions necessary to replicate these results have been deposited at Zenodo, doi:10.5281/zenodo.5873982. IDBac is freely available from github.com/chasemc/IDBacApp and doi:10.5281/zenodo.1115619. All MALDI-TOF MS spectra have been deposited in MASSIVE: MSV000089059; doi:10.25345/C5DN40009.

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
