# Peer review of "Automated Microbial Library Generation Using the Bioinformatics Platform IDBac"

_molecules, 2022, doi:10.3390/molecules27072038_

Round 1
Reviewer 1 Report
After careful evaluation of manuscript title"Automated microbial library generation using the bioinformat-2 ics platform IDBac" pleased to accept the manuscript for publication in its current form.Author Response
No response necessary.
Reviewer 2 Report
The report entitled “Automated microbial library generation using the bioinformatics platform IDBac” describes a new algorithm for prioritizing isolates without human bias on the ID-Bac platform for solving redundancy in microbial screening. This paper is also valuable in reducing various problems in microbial screening, such as time, effort, cost, and so on. I would like authors to consider the following point.
- Please discuss what people think is important for manual curation, and what only the algorithm is "seeing". I think it will be useful for future algorithm development.
- Is it possible to reflect the amount of NP produced by the isolated on this platform?
Author Response
1. The algorithm and the manual curator have access to the same dataset, so they are both analyzing the same data. In regard to manual curation, this decision is left to the individual researcher and should be tailored specifically to their program. We are not in a position to comment on the individual needs of different programs.
2. Unfortunately no, this is a limitation of MALDI MS/IDBac, though this is also out of the scope of what the method was intended to be.
Reviewer 3 Report
This communication prepared by Clark et al. provides an important update to the NP community regarding the tool, IDBac, developed by the same authors. The move towards automation will surely be greatly appreciated by the community (it would be nice in the article to list usage thus far of the tool if that data exists). The additions to the IDBac tool have been clearly presented and the comparison between the automated workflow and the manual interpretation are effective and clear as well. I’m under the impression that most high-throughput workflows have now integrated some level of automated dereplication that should help account for redundancies at early stages. However I think a tool like IDBac identifies some of these redundancies at the earliest stage and also represents a highly cost effective (time and money) technique that more labs could potentially facilitate given the right equipment. I appreciate the candidness in mentioning the manual aspects still present in the workflow and this awareness clearly shows where you all plan to take IDBac in the future. I do believe this could be easily integrated into a multitude of a high-throughput workflows, especially for organizations with vast culture collections that might not ever be fully explored in a single researcher’s lifetime.
My one question and maybe this is less a problem and more a discussion to have overall. A central caveat with MS-based metabolomics of microbial natural products is how much of the overall secondary metabolism is one capturing in order to evaluate the potential of a strain. The point I am trying to get at is with your evaluation of secondary metabolism using one media type, are you truly capturing enough of the chemical space to evaluate which microbes might be more talented or interesting than others via number of unique features? Also since you are taking colony material, are you missing diffused metabolites? Since no biological screening is utilized as an indicator of biologically active potential, potential chemical novelty is the indicator (which I think is actually a good thing to try to find true chemical novelty and not simply analogs or known metabolites).
In the original IDBac paper, the authors indicate the relevance of microbial growth on various media, conditions, etc. They state that for proteomics, this is negligible but as we all understand regarding OSMAC and metabolomics, this is widely variable. Naturally, it is impractical for any one study to vigorous examine even 3 conditions for so many isolates but I do wonder if unintentionally the resulting output may falsely bias those ‘stressed’ strains or ones triggered by the media used in the study. Coupling genomics to this type of data would actually see if strains with higher chemical potential actually correlated to biosynthetically talented bacteria. Ideally IDBac would integrate/collate data from different media types into a single microbial metabolome (this is clearly not for this publication and I can see it being smoothly integrated into future iterations of IDBac). This would generate a more robust dataset covering more chemical space and thusly predictions may continue to improve.
Overall, a pleasure to read this update and it’s great to see community-based tools continue to be developed.
Author Response
We thank Reviewer 3 for a very thorough report and valuable suggestions of how to improve IDBac. They have a strong understanding of the strengths and limitations of the tool. Our thoughts and actions are listed below.
- "it would be nice in the article to list usage thus far of the tool if that data exists". We input usage statistics on p2 l52.
- "The point I am trying to get at is with your evaluation of secondary metabolism using one media type, are you truly capturing enough of the chemical space to evaluate which microbes might be more talented or interesting than others via number of unique features?" This is an excellent point and one we have struggled with for years since IDBac's adoption. Generally speaking, we suspect that we are capturing enough chemical space in order to make informed decisions. However the Reviewer is correct, in some cases it is possible that many of the specialized metabolites that make a strain unique may not be expressed under the given set of culture conditions. The user can account for this by cultivating an individual colony under multiple growth conditions and compiling the data in IDBac. Of course this would more than double the workload but it is still possible within the IDBac system. To truly answer the reviewer's question we would need to couple whole genome data of known strains and compare them with IDBac metabolite association networks, accounting for each gene cluster that is present in a strain and prodices a metabolite. We would then have to assess multiple different microbial taxa and assess ~ what percent of clusters are represented in IDBac data. Of course, this would be a huge undertaking that would require many scientists and years of work (and is out of the scope of this manuscript).
- "Also since you are taking colony material, are you missing diffused metabolites?" This is an excellent question. From our past experiences, most diffused metabolites can also be detected directly from the cell material as well. Of course, it is entirely possible that a metabolite is only detectable in the agar and not the cell mass. This is a limitation of the method.